# Reliability and Validity of the Modified Korean Version of the Chalder Fatigue Scale (mKCFQ11)

**DOI:** 10.3390/healthcare8040427

**Published:** 2020-10-24

**Authors:** Yo-Chan Ahn, Jin-Seok Lee, Chang-Gue Son

**Affiliations:** 1Department of Health Service Management, Daejeon University, Daejeon 300-716, Korea; ycahn@dju.kr; 2Department of Korean Medicine, Institute of Bioscience and Integrative Medicine, Daejeon University, Daejeon 300-716, Korea; neptune@dju.ac.kr

**Keywords:** chronic fatigue syndrome, chalder fatigue scale, visual analogue scale, fatigue severity scale

## Abstract

Fatigue can accompany various diseases; however, fatigue itself is a key symptom for patients with chronic fatigue syndrome (CFS). Due to the absence of biological parameters for the diagnosis and severity of CFS, the assessment tool for the degree of fatigue is very important. This study aims to verify the reliability and validity of the modified Korean version of the Chalder Fatigue Scale (mKCFQ11). This study was performed using data from 97 participants (Male: 37, Female: 60) enrolled in a clinical trial for an intervention of CFS. The analyses of the coefficient between the mKCFQ11 score and the Fatigue Severity Scale (FSS), the Visual Analogue Scale (VAS) or the 36-item Short-Form Health Survey (SF-36) at two time points (baseline and 12 weeks) as well as their changed values were conducted. The mKCFQ11 showed strong reliability, as evidenced by the Cronbach’s alpha coefficient of 0.967 for the whole item and two subclasses (0.963 for physical and 0.958 for mental fatigue) along with the suitable validity of the mKCFQ11 structure shown by the principal component analysis. The mKCFQ11 scores also strongly correlated (higher than 0.7) with the VAS, FSS and SF-36 on all data from baseline and 12 weeks and changed values. This study demonstrated the clinical usefulness of the mKCFQ11 instrument, particularly in assessing the severity of fatigue and the evaluation of treatments for patients suffering from CFS.

## 1. Introduction

Fatigue is a subjective complaint commonly experienced by the general population during their lifetimes, with an approximate 30–50% point prevalence [1]. Unlike acute fatigue, which disappears after resting or treatment of the causative diseases, uncontrolled chronic fatigue substantially impairs the health-related quality of life [2]. In particular, chronic fatigue syndrome (CFS), a typical medically unexplained chronic fatigue, is a debilitating illness that results in the unemployment of half of patients with CFS and a risk of suicide approximately seven-fold higher than that of healthy controls [3,4].

Although many findings have been achieved from diverse aspects, including the nervous system, endocrine system, immune system, metabolomics, and gut microbiota, no universally accepted etiology, pathophysiology, diagnosis, or treatment for CFS exists [5]. Accordingly, both physicians and patients encounter many difficulties in the management of this disorder and communication with each other [6]. The diagnosis of certain disorders and the assessment of their severity are fundamental steps in treatment processes. For the diagnosis of CFS, physicians have adapted case definitions or diagnostic criteria such as the Fukuda definition in 1994 by the Centers for Disease Control and Prevention (CDC) or the criteria by Institute of Medicine (IOM) in 2015 [7,8]. These tools have been developed depending upon the clinical features.

In general, objective measurement of fatigue is very important for patient management as well as assessment of the intervention efficacy for fatigue-related disorders [9]. Since the diagnosis of CFS and its categorization of illness status rely on self-reporting consultation, an accurate quantification of fatigue severity and its associated symptoms is vital, especially for patients with CFS [10]. To date, many severity scales have been developed based on patient-reported outcomes (PROs) for CFS patients, such as the Visual Analogue Scale (VAS), Fatigue Severity Scale (FSS), Multidimensional Fatigue Inventory (MFI), Chalder Fatigue Scale (CFQ), and Fibromyalgia and Chronic Fatigue Syndrome Rating Scale (FibroFatigue scale) [11,12,13]. Most of these instruments, however, have limitations, including a lack of specificity for CFS compared to other fatigue-inducing disorders, such as primary depression [14].

Among many fatigue-measuring scales, one of the most commonly used is the CFQ, which was developed in 1993 [12]. The CFQ consists of 11 easily applicable self-rating items for two domains of physical and mental fatigue, which could clearly discriminate patients with CFS from a healthy control [15]. This instrument has been well adapted in studies not only for clinical features of patients with CFS but also for evaluations of interventions, such as rehabilitative therapies [16,17]. The Korean version of the CFQ (K-CFQ) was also validated using healthy subjects, Korean graduate students [18]. The CFQ was initially developed as a four-point scale that compares to the “usual” status, and thus, it is difficult to measure the change in fatigue severity for certain periods. Therefore, we slightly modified it into a 10-point Likert scale (between normal and worst status) to describe their illness condition after treatment, called the Modified Korean Version of the Chalder Fatigue Scale (mKCFQ11). We have adapted the mKCFQ11 as a primary measurement in clinical trials using herb-derived therapeutics in both patients with idiopathic chronic fatigue (ICF) and CFS [19,20].

Although this modified scale has been well applied, its reliability and validity have not yet been assessed. The present study thus aims to verify the reliability and validity of the mKCFQ11.

## 2. Materials and Methods 

### 2.1. Study Population and Data Source 

The study population comprised 97 patients with CFS (37 males and 60 female) between the ages of 18 and 65 years (mean age 39.6 ± 10.0 years) who were enrolled in a phase 2 trial conducted in two hospitals (Daejeon Korean Medicine Hospital of Daejeon University and Daejeon St. Mary’s Hospital of the Catholic University of South Korea) from December 2016 to November 2017 [20]. All participants met the 1994 Fukuda CFS definition, which requires clinically evaluated, unexplained, persistent, or relapsing chronic fatigue [7]. The exclusion criteria were subjects who suffered from other illnesses that induced chronic fatigue within the past 6 months, such as anemia, liver, kidney, and thyroid dysfunction, depression and anxiety disorders.

The data resource for this validation study was from the above phase 2, randomized, placebo-controlled trial of Myelophil (a standardized anti-fatigue herbal agent). This trial was designed to primarily determine the efficacy of Myelophil using the changes in the CFQ-based fatigue scores between baseline and 12 weeks of treatment. In addition, this trial had another purpose to verify the reliability and validity of the mKCFQ11 for use in trial 3; thus, we used the mKCFQ11 data at two time points and its changes regardless of the allocation of participants. Two well-known fatigue instruments, the VAS and the FSS, and the 36-item short-form health survey (SF-36) as an indicator of health-related quality of life (QoL) were used to calculate Pearson correlation coefficients for the mKCFQ11.

### 2.2. Ethics Statement 

The trial was implemented in accordance with ethical and safety guidelines upon the approval of the Ministry of Food and Drug Safety (MFDS) in South Korea (Approval number 12354) and the Institutional Review Board in two hospitals (approval number DJDSKH-17-DR-03 in Daejeon Korean Medicine Hospital, DIRB-00139-3 in Daejeon St. Mary’s Hospital). This trial is registered at Clinical Research Information Service (CRIS) in Korea with identifier number KCT0002317, and an independent medical monitor (by MEDICAL excellence) ensured the trial procedure according to the protocol and maintaining the data. Data were independently analyzed by a medical statistics specialist.

### 2.3. Modified Korean Version of CFQ (mKCFQ11)

The original version of the CFQ consists of 11 items to determine the fatigue-related status by comparing to the “usual” condition: “Less than usual”, “No more than usual”, “More than usual”, and “Much more than usual”. However, the reference point (“usual”) made it difficult for Korean patients to express their illness status, especially for patients with CFS due to the long-term duration of this condition, which could be over 10 years, or the very frequent childhood diagnosis. Furthermore, this “usual”-based comparison of illness condition at certain time points was not easily adapted to measure the changed score of fatigue severity in clinical trials of intervention. Therefore, we slightly modified it into a 10-point Likert scale as (0 = not at all to 9 = unbearably severe condition) for the same 11 questions (physical fatigue questions 1st–7th items, and mental fatigue 8th–11th items, total score range 0–99).

Briefly, the English version of the CFQ11 questionnaire was independently translated into Korean by two Korean specialists on CFS and a native English speaker proficient in Korean. Next, four specialists reviewed the differences and merged them and then examined the practical performance of many patients suffering from fatigue, including CFS. Based on the responder’s comments, specialists discussed and completed the Korean version of the CFQ with 10-point Likert scale. After repeated tests of the patients complaining of chronic fatigue, including CFS, and a pilot clinical trial for ICF [19], the final mKCFQ11 (Table 1) was determined and used as a primary measurement for the above trial [20].

### 2.4. Fatigue Severity Scale (FSS)

The FSS is a 9-item self-report questionnaire to easily measure physical, social, or cognitive effects of fatigue. This scale was developed in 1989 as a seven-point Likert scale (1 indicating “Strongly disagree” to 7 representing “Strongly agree”, total score range 7–63) and was translated into Korean previously and shown to be clinically useful for patients with fatigue [21,22].

### 2.5. Visual Analogue Scale (VAS)

The VAS was assessed by asking the participants to specify their level of overall discomfort from CFS by indicating a position along a continuous 100 mm line between two end points. The left end indicated “no exhaustion at all” while the right end indicated “complete exhaustion”, and the value was then determined by measuring the length (mm) from the left end of the line [23].

### 2.6. The 36-Item Short-Form Health Survey (SF-36)

Health-related quality of life was measured using a 36-item short-form health survey (SF-36), whose usefulness was confirmed in patients with CFS [24] and translated into the Korean version [25]. The SF-36 consists of eight scaled scores that broadly reflect two domains of physical and mental health status. The total score range of each domain is a minimum of 0 (indicating “maximum disability”) to a maximum of 100, representing “no disability”.

### 2.7. Statistical Analysis

All statistical analyses were performed using Predictive Analytics SoftWare (PASW) Statistics (SPSS Inc., Chicago, IL, USA) for Windows. Two domains of mKCFQ11 (physical and mental) were assessed for their internal consistency by using Cronbach’s alpha. Principal component analysis (PCA) was performed to examine the mKCFQ11 factor structure; factors with eigen values of >1 was extracted. The convergent validity of the total cognition score was tested using the Pearson correlation coefficients with the FSS, VAS and SF-36. Differences between the fatigue groups were tested using the t-test and one-way analysis of variance (ANOVA).

## 3. Discussion

The present study aimed to verify the reliability and validity of the mKCFQ11, a modified Korean version of the Chalder Fatigue Scale, altered from a four-point scale comparing “usual” status to a 10-point Likert scale (between normal and worst status) was created. This modification was performed because Korean patients with CFS described the difficulty of assessing their fatigue-associated severity using the CFQ instrument, especially in comparison to the “usual”. In fact, most patients with CFS have been diagnosed with the disease for many years of disease with fluctuating symptoms [26]; thus, they frequently hesitated in answering. The mKCFQ11 has been adapted well by participants, in particular RCTs because they chose to describe their condition between “no fatigue” and ”unbearable fatigue” at a certain period [20].

The reliability of the mKCFQ11 was strongly shown from the results using Cronbach’s alpha coefficients. In general, Cronbach’s alpha values ≥ 0.7 are considered satisfactory [27], and the values of mKCFQ11 were 0.967 for total fatigue and 0.963 and 0.958 for the subscales of physical and mental fatigue. These internal consistencies of the mKCFQ11 were higher than those of the original English version of the CFQ (total value of 0.92) using 361 patients with CFS in England [15] or the K-CFQ using Korean graduate students (total value was 0.88, 0.87 for physical and 0.73 for mental fatigue) [18]. There must be differences in not only the language structure between English and Korean but also cultural gaps, affecting the final results after translations of self-report measures [28]. When we translated the CFQ, we therefore strictly reflected the initial meaning of each question but tried to make them easy for the patients to understand via modification of the English phrases.

The principal component analysis using the varimax rotation model demonstrated the suitable validity of the mKCFQ1 structure composed of 11 question items with two subclasses of physical and mental fatigue (Figure 1 and Table 1 and Table 2). In addition, the mKCFQ11 was significantly correlated with the VAS, FSS and SF-36 on three different data points from baseline and 12 weeks (Table 3). These correlations were stronger at 12 weeks than at baseline, as anticipated, because the intervention (Myelophil) showed positive effects on all scores of the mKCFQ11, FSS, VAS and SF-36 compared to the control [20]. Furthermore, these strong correlations were repeated for the altered values of the mKCFQ11 and others. The correlations between mKCFQ11 (total and subclasses of physical and mental fatigue) and the FSS or VAS were higher than 0.7, which indicated the very strong associations between the two fatigue scales [29]. FSS is a well-known fatigue measure that is specific for individuals with CFS compared to those with multiple sclerosis or primary depression [30]. Among PRO-based measurements, VAS is a simple technique to obtain continuous- and interval-level measurement data and to reduce response-style biases of Likert-type scales [31]. Moreover, the correlation with the SF-36 was relatively low compared to that with the FSS or VAS, which could be because the SF-36 is a non-disease specific generic scale to assess health-related quality of life [32]. The RCT for Myelophil measured the serum concentrations for oxidative and antioxidant parameters, and then the statistically significant correlations with mKCFQ11 were observed only for total glutathione (GSH) contents, tumor necrosis factor-α (TNF-α) and interferon-gamma (IFN-γ) on baseline and for the changed values of reactive oxygen species (ROS) after 12 weeks of treatment (Appendix A). 

In the assessment of subjective complaint disorders such as CFS or chronic pain, disease-specific PRO measures are the most important strategy to determine the treatment response because the patient is the most important judge of whether changes are important or meaningful [33,34]. The Multidimensional Fatigue Inventory (MFI-20) is another typical instrument used to assess fatigue severity in individuals with CFS, and the reliability and validity of its Korean version (MFI-K) was recently compared to the VAS and FSS [35]. The Cronbach’s alpha coefficient of the MFI-K was 0.88, and the correlation coefficients with the VAS score (0.419) and the FSS score (0.635) were lower than that of the mKCFQ11. On the other hand, regarding the appropriate selection of the participants with CFS for especially RCTs, we would like to now recommend the combination of any diagnostic instrument such as CDC 1994 or IOM diagnostic criteria and cut-off scores of severity using mKCFQ11 or MFI.

## 4. Results

### 4.1. General Characteristics

A total of 97 participants (37 males and 60 females) who had a median age of 40 years (range 21 to 64 years) and a mean body mass index (BMI) of 22.6 ± 2.6 were included. All measurement scores for fatigue severity showed improvements at 12 weeks compared to at zero weeks, such as from 61.9 ± 15.5 to 37.7 ± 17.9 in the mKCFQ11, from 7.1 ± 1.7 to 4.3 ± 2.0 in the VAS, from 45.4 ± 9.8 to 32.3 ± 12.0 in the FSS, and 89.8 ± 15.8 to 101.2 ± 13.2 in the SF-36 (Table 4). These results were expected because nearly half of the participants (48 of 97) had taken an anti-fatigue herbal agent (Myelophil) for 12 weeks. There was no gender-related difference in score of mKCFQ11 (data not shown).

### 4.2. Internal Consistency

The Cronbach’s alpha coefficient of mKCFQ11 was 0.967. The internal consistencies of the two subclasses were as follows: 0.963 for physical fatigue (Q1 to Q7) and 0.958 for mental fatigue (Q8 to Q11). In the absence condition for each item, Cronbach’s alpha values were smaller than the values by the all-existence condition, which indicates the internal consistency of all question items.

### 4.3. Structural Validity of the mKCFQ11

The result of principal component analysis (with a varimax rotation) showed the structural validity of the mKCFQ11. Each question item was clustered together according to two subclasses and was distinct from the other item (Figure 1 and Table 1). Two subclass factors explained 85.3% of the total variance. Factor one explained 75.6% of the total variance and included all of the “Physical fatigue” items (Q1 to Q7). Factor two explained 9.7% of the total variance and included all “Mental fatigue” items (Q8 to Q11) (Table 2)

### 4.4. Convergent Validity with the VAS, FSS and SF-36

The mKCFQ11 had good convergent validity. The total mKCFQ11 score was significantly correlated with the VAS, FSS and SF-36 at both baseline (zero weeks, *p* < 0.001) and 12 weeks (*p* < 0.001). In addition, two subclasses (physical and mental fatigue) scores were also well correlated with the scores, with statistical significance at both timepoints (except between mental fatigue and mental SF-36 score at zero weeks, Table 3).

### 4.5. Convergent Validity of Changed Values with the VAS, FSS and SF-36

The changed values of the mKCFQ11 between zero weeks and 12 weeks were also significantly correlated with those of VAS, FSS and SF-36 (*p* < 0.001). Moreover, the changes in the two subclasses (physical and mental fatigue) scores were also significantly correlated with the VAS, FSS and SF-36 scores (physical and mental SF-36, Table 3).

## 5. Conclusions

This study demonstrated the clinical usefulness of the mKCFQ11, particularly in assessing the degree of fatigue and the changes in fatigue-related symptoms after treatment of patients with CFS. However, further studies are required, especially regarding subjects with other fatigue disorders and comparisons with an unmodified version of the CFQ.

## Figures and Tables

**Figure 1 healthcare-08-00427-f001:**
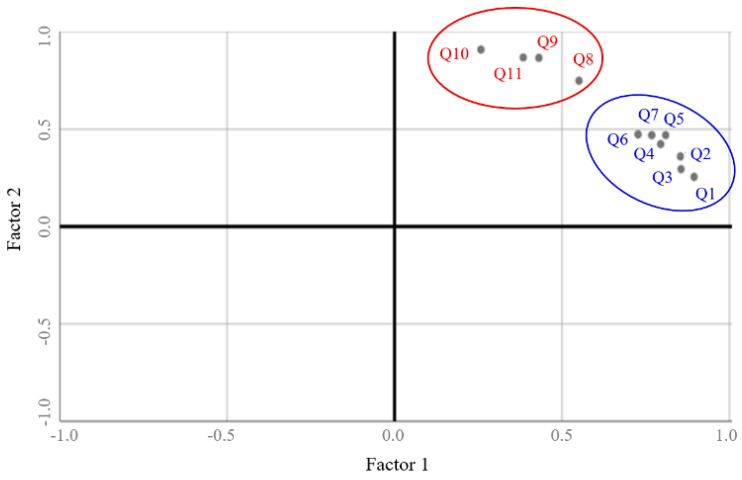
Principal component analysis (PCA) of the mCFQ11 structure. PCA was conducted for two-Figure 1. Q1 to Q7) and mental fatigue (factor 2, Q8 to Q11) based on covariance for a scale reliability of 11 questions.

**Table 1 healthcare-08-00427-t001:** The mKCFQ11 and its component analysis.

Korean Questionnaire	Varimax Rotation	
	Factor 1	Factor 2
1. 당신이 평소 느끼는 피로의 정도는 어떻습니까? (Do you have problems with tiredness?)	0.852	0.255
2. 당신은 어느 정도의 휴식이 필요합니까? (Do you need to rest more?)	0.859	0.294
3. 당신은 어느 정도의 졸음을 느끼십니까? (Do you feel sleepy or drowsy?)	0.854	0.360
4. 당신은 피로감 때문에 일을 시작할 때 힘이 듭니까? (Do you have problems starting things?)	0.795	0.423
5. 당신은 기운(기력)이 없다고 느끼십니까? (Do you lack energy?)	0.810	0.469
6. 당신은 근육의 힘이 약해졌다고 느끼십니까? (Do you have less strength in your muscles?)	0.727	0.473
7. 당신은 허약해졌다고 느끼십니까? (Do you feel weak?)	0.768	0.469
8. 당신은 일에 대한 집중력이 떨어졌습니까? (Do you have difficulties concentrating?)	0.551	0.748
9. 당신은 명료하게 생각하는 것에 어려움이 있습니까? (Do you make slips of the tongue when speaking?)	0.431	0.866
10. 당신은 말할 때 적절한 단어선택이 어려운 경우가 있습니까? (Do you find it more difficult to find the right word?)	0.258	0.908
11. 당신의 기억력 저하는 없습니까? (How is your memory?)	0.384	0.868

The modified Korean Version of the Chalder Fatigue Scale (mKCFQ11) is 10-point Likert scale (0 = ‘not at all’ to 9 = ‘unbearably severe condition’), while Chalder Fatigue Scale (CFQ) is 4-point scale (‘less than usual’, ‘no more than usual’, ‘more than usual’ and ‘much more than usual’ for Q1 to Q10 and ‘better than usual’, ‘no worse than usual’, ‘worse than usual’, and ‘much worse than usual’ for Q10).

**Table 2 healthcare-08-00427-t002:** Principal component analysis after varimax rotation of the mKCFQ11.

N. of Subclass	Variance Explained			Extraction of Sums of Squared Leading		
	Eigenvalue	% Variance	% Cumulative	Factor	Subclass Factor 1	Subclass Factor 2
1	8.3126	75.597	75.597	Total	8.32	1.07
2	1.066	9.692	85.288			
3	0.387	3.514	88.803	% total of variance	75.6	9.7
4	0.261	2.369	91.172			
5	0.212	1.929	93.100	Questions	Q1 Q2 Q3 Q4 Q5 Q6 Q7	Q8 Q9 Q10 Q11
6	0.189	1.716	94.816			
7	0.171	1.551	96.368			
8	0.139	1.267	97.634			
9	0.117	1.067	98.701			
10	0.083	0.751	99.453	Interpretation	Physical	Mental
11	0.060	0.547	100.000			

After varimax rotation of mKCFQ11, the number of subclasses was determined as initial Eigenvalue > 1.

**Table 3 healthcare-08-00427-t003:** Correlation between the mKCFQ11 and the FSS, VAS or SF-36 on 0 and 12-week.

mKCFQ11		FSS	VAS	SF-36
0-week	Total	0.755 **	0.732 **	−0.383 **
	Physical	0.766 **	0.777 **	−0.403 **
	Mental	0.659 **	0.608 **	−0.311 **
12-week	Total	0.859 **	0.862 **	−0.698 **
	Physical	0.825 **	0.864 **	−0.672 **
	Mental	0.812 **	0.760 **	−0.672 **
Changes between 0 and 12-week	Total	0.757 **	0.860 **	−0.592 **
	Physical	0.769 **	0.887 **	−0.596 **
	Mental	0.698 **	0.761 **	−0.527 **

The mKCFQ11: Modified Korean Version of the Chalder Fatigue Scale, VAS: Visual Analogue Scale, FSS: Fatigue Severity Scale, SF-36: 36-item Short-Form Health Survey. The statistical significance of correlation was presented as ** *p* < 0.001.

**Table 4 healthcare-08-00427-t004:** Characteristics of the subjects and measurements.

Basic Characteristic	Male	Female	Total
Number of subjects (%)	37 (37.8)	60 (62.8)	97 (100)
Median age (year, range)	42 (25 to 63)	39 (21 to 64)	40 (21 to 64)
Mean value of BMI (kg/m^2^)	24.1 ± 1.9	21.7 ± 2.6	22.6 ± 2.6
Measurement	0-week	12-week	Change
mKCFQ11: Total	61.9 ± 15.5	37.7 ± 17.9	24.2 ± 20.5
Physical	42.3 ± 9.1	26.0 ± 11.7	16.3 ± 13.0
Mental	19.9 ± 7.3	11.9 ± 6.8	8.0 ± 8.2
VAS	7.1 ± 1.7	4.3 ± 2.0	2.8 ± 2.4
FSS	45.4 ± 9.8	32.3 ± 12.0	13.2 ± 13.2
SF-36	89.8 ± 15.8	101.2 ± 13.2	−11.4 ± 17.2

BMI: Body Mass Index, mKCFQ11: Modified Korean Version of the Chalder Fatigue Scale, VAS: Visual Analogue Scale, FSS: Fatigue Severity Scale, SF-36: 36-item Short-Form Health Survey.

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
