# Peer review of "Reliability and Validity of the Modified Korean Version of the Chalder Fatigue Scale (mKCFQ11)"

_healthcare, 2020, doi:10.3390/healthcare8040427_

Round 1

Reviewer 1 Report

Line 29 lifetime[s]

Agreement error

Line 39 In general, the diagnosis of certain disorders and the assessment  of their severity are fundamental steps; then, the diagnosis of CFS has adapted case definitions or diagnostic criteria developed depending upon the clinical features, such as the Fukuda definition in 1994 by Disease Control and Prevention (CDC) and recent criteria by the Institute of Medicine (IOM) in 2015 [7,8].

Not clear. Advise simplify and rewrite

Reference #20 I read the 2019 Myelophil efficacy study which raised a couple of questions regarding the mKCFQ11. Might these be usefully explored in the present paper?

Parameters of Oxidative Stress and Cytokines

The serum samples from 97 participants were analyzed before and after treatment. There was no significant difference in the levels of oxidative indicators (serum reactive oxygen species and malondialdehyde), antioxidative indicators (total antioxidant capacity, catalase, superoxide dismutase, total GSH, GSH-Px, and GSH-Rd), and cytokines (TNF-α and IFN-γ) between the two groups. A post hoc subgroup analysis also did not show any significant differences. Details are shown in Supplementary Table 4.

Did you correlate mKCFQ11 scores with biological measures? If so, could you include results and discuss in the present paper?

We have speculated on the potential reasons why Myelophil did not show a significant therapeutic effect. First, a high placebo effect can mask the efficacy of Myelophil. Our study showed a high placebo response rate of over 30%, whereas a previous meta-analysis reported that the placebo response rate of CFS patients was, on average, 20% (Cho et al., 2005). Many Korean people believe that traditional herbal medicines are beneficial for fatigue, which attributed to the high placebo effect in this study. Second, patient bias may have occurred. We enrolled the participants by the CDC criteria based on their answers to the yes/no questions. Since these criteria could not measure the severity of patients’ illnesses, the patient population may have included participants with mild levels of symptoms.

Accordingly, we need to consider the MFI as the primary end point and participants with moderate and severe symptoms of CFS in the next phase 3 trial with Myelophil.

Does this suggest that the mKCFQ11 may not discriminate among all patients diagnosed with ME/CFS? Is this a limitation that should be addressed in the present paper?

NB It is our experience that while SF36 does discriminate between ME/CFS and other conditions it is less useful for comparing symptom severity among ME/CFS patients. Is this also an issue with the mKCFQ11?

Author Response

Line 29 lifetime[s]

  • We appreciate reviewer for the correction, and we revised the word.

Agreement error

  • According to the indication of reviewer, we revised the sentence.

Line 39 In general, the diagnosis of certain disorders and the assessment of their severity are fundamental steps; then, the diagnosis of CFS has adapted case definitions or diagnostic criteria developed depending upon the clinical features, such as the Fukuda definition in 1994 by Disease Control and Prevention (CDC) and recent criteria by the Institute of Medicine (IOM) in 2015 [7,8].

Not clear. Advise simplify and rewrite

  • We thank reviewer for the helpful comment. Werewrite the sentence to simplify it.

Reference #20 I read the 2019 Myelophil efficacy study which raised a couple of questions regarding the mKCFQ11. Might these be usefully explored in the present paper?

  • We appreciate reviewer for the professional question. We believe that those questions have been explored in the present paper through the following answers.

Parameters of Oxidative Stress and Cytokines

The serum samples from 97 participants were analyzed before and after treatment. There was no significant difference in the levels of oxidative indicators (serum reactive oxygen species and malondialdehyde), antioxidative indicators (total antioxidant capacity, catalase, superoxide dismutase, total GSH, GSH-Px, and GSH-Rd), and cytokines (TNF-α and IFN-γ) between the two groups. A post hoc subgroup analysis also did not show any significant differences. Details are shown in Supplementary Table 4.

Did you correlate mKCFQ11 scores with biological measures? If so, could you include results and discuss in the present paper?

  • According to reviewer’ professional recommendation, we re-analyzed the correlation between mKCFQ11 scores and biological measures. We added this information in ‘Discussion’ section and Supplementary Table 1.

We have speculated on the potential reasons why Myelophil did not show a significant therapeutic effect. First, a high placebo effect can mask the efficacy of Myelophil. Our study showed a high placebo response rate of over 30%, whereas a previous meta-analysis reported that the placebo response rate of CFS patients was, on average, 20% (Cho et al., 2005). Many Korean people believe that traditional herbal medicines are beneficial for fatigue, which attributed to the high placebo effect in this study. Second, patient bias may have occurred. We enrolled the participants by the CDC criteria based on their answers to the yes/no questions. Since these criteria could not measure the severity of patients’ illnesses, the patient population may have included participants with mild levels of symptoms.

Accordingly, we need to consider the MFI as the primary end point and participants with moderate and severe symptoms of CFS in the next phase 3 trial with Myelophil.

Does this suggest that the mKCFQ11 may not discriminate among all patients diagnosed with ME/CFS? Is this a limitation that should be addressed in the present paper?

  • We really appreciate reviewer for the professional comments and question. We now recognized that the combination of any diagnostic instrument (CDC 1994 or IOM diagnostic criteria) and cut-off scores of severity (using mKCFQ11 or MFI) may be strongly recommend to select the suitable participants with CFS in especially RCTs.

NB It is our experience that while SF36 does discriminate between ME/CFS and other conditions it is less useful for comparing symptom severity among ME/CFS patients. Is this also an issue with the mKCFQ11?

  • We thank reviewer for the professional comment and question. We are also sure that SF36 cannot discriminate between ME/CFS and other conditions. Regarding mKCFQ11, it is much useful (even comparing to CFQ11) for comparing symptom severity among ME/CFS patients, and then mKCFQ11 would be able to adapted as a main assessment tool in clinical practices and RCTs for CFS.

Reviewer 2 Report

In this manuscript, authors developed Korean version of the CFS assessment tool revising Chalder Fatigue Scale, which is widely used in the field. Authors well translated the Chalder Fatigue Scale to Korean version and well validated the tool in this study. This tool would be useful in Korea.

Minor, it would be informative, if authors report/discuss splitting the 12wk-herval-fatigue improvement results by sex.

Author Response

Comments and Suggestions for Authors

In this manuscript, authors developed Korean version of the CFS assessment tool revising Chalder Fatigue Scale, which is widely used in the field. Authors well translated the Chalder Fatigue Scale to Korean version and well validated the tool in this study. This tool would be useful in Korea.

Minor, it would be informative, if authors report/discuss splitting the 12wk-herval-fatigue improvement results by sex

  • We thank reviewer for the professional suggestion. When we analyzed the effects of Myelophil on CFS, there was no gender-related difference in score of mKCFQ11. We added this information in the present revised manuscript.